# Genetic Background Influences Severity of Colonic Aganglionosis and Response to GDNF Enemas in the *Holstein* Mouse Model of Hirschsprung Disease

**DOI:** 10.3390/ijms222313140

**Published:** 2021-12-05

**Authors:** Rodolphe Soret, Nejia Lassoued, Grégoire Bonnamour, Guillaume Bernas, Aurélie Barbe, Mélanie Pelletier, Manon Aichi, Nicolas Pilon

**Affiliations:** 1Molecular Genetics of Development Laboratory, Département des Sciences Biologiques, Université du Québec à Montréal (UQAM), Montréal, QC H3C 3P8, Canada; rode440@gmail.com (R.S.); nejia_lassoued11@yahoo.fr (N.L.); bonnamour.gregoire@courrier.uqam.ca (G.B.); guillaume.bernas@gmail.com (G.B.); aureliebarbe0@gmail.com (A.B.); pelletier.melanie@courrier.uqam.ca (M.P.); manon.chichi@hotmail.fr (M.A.); 2Centre D’excellence en Recherche sur les Maladies Orphelines—Fondation Courtois (CERMO-FC), Université du Québec à Montréal, Montréal, QC H2X 3Y7, Canada; 3Département de Pédiatrie, Université de Montréal, Montréal, QC H3T 1C5, Canada

**Keywords:** enteric nervous system, GDNF, genetic background, Hirschsprung disease, melanocytes, mouse model, neural crest, pigmentation, regenerative medicine, tissue-resident stem cells

## Abstract

Hirschsprung disease is a congenital malformation where ganglia of the neural crest-derived enteric nervous system are missing over varying lengths of the distal gastrointestinal tract. This complex genetic condition involves both rare and common variants in dozens of genes, many of which have been functionally validated in animal models. Modifier loci present in the genetic background are also believed to influence disease penetrance and severity, but this has not been frequently tested in animal models. Here, we addressed this question using *Holstein* mice in which aganglionosis is due to excessive deposition of collagen VI around the developing enteric nervous system, thereby allowing us to model trisomy 21-associated Hirschsprung disease. We also asked whether the genetic background might influence the response of *Holstein* mice to GDNF enemas, which we recently showed to have regenerative properties for the missing enteric nervous system. Compared to *Holstein* mice in their original FVB/N genetic background, *Holstein* mice maintained in a C57BL/6N background were found to have a less severe enteric nervous system defect and to be more responsive to GDNF enemas. This change of genetic background had a positive impact on the enteric nervous system only, leaving the neural crest-related pigmentation phenotype of *Holstein* mice unaffected. Taken together with other similar studies, these results are thus consistent with the notion that the enteric nervous system is more sensitive to genetic background changes than other neural crest derivatives.

## 1. Introduction

Hirschsprung disease (HSCR) is a birth defect of the neurocristopathy class characterized by the lack of enteric neural ganglia (aganglionosis) in the distal bowel [1,2]. This occurs because neural crest-derived progenitors of the enteric nervous system (ENS) have failed to complete their colonization of the gastrointestinal tract during prenatal development [3]. In absence of ENS innervation, smooth muscles from the affected distal bowel segment remain tonically contracted, causing a functional intestinal obstruction (megacolon). Because the ENS influences epithelial barrier function as well, children with HSCR also have a high risk of enterocolitis leading to sepsis and premature death [4,5]. 

HSCR is a complex genetic condition characterized by non-Mendelian inheritance [1,6,7], as best exemplified by variation in disease penetrance and severity between family members carrying a similar set of mutations [8,9,10]. Current knowledge suggests that most cases are due to different combinations of rare coding variants, common regulatory variants, and/or copy-number variants in many genes important for proper colonization of the developing gastrointestinal tract by ENS progenitors [11,12]. Dozens of such genes have been identified so far [11,12,13,14,15,16,17,18,19]. The major HSCR-associated gene is *RET* (REarranged during Transfection) [20,21], which codes for a transmembrane tyrosine kinase that is activated by GDNF (Glial cell line-derived Neurotrophic Factor) upon binding to the co-receptor GFRα1 (GDNF family receptor alpha-1). Other genes that influence HSCR risk include transcription factors (*SOX10, ZFHX1B, PHOX2B*), EDNRB signaling pathway molecules (*EDNRB, EDN3, ECE1*), cell adhesion proteins (*L1CAM*), guidance molecules (*SEMA3D*), molecules needed for cell-extracellular matrix interactions (*ITGB1* and *COL6*), and diverse additional genes (*BACE2, NRG1, ERBB2, ADAMTS17, ACSS2*) [11,12]. Male sex and trisomy 21 are other contributing factors, increasing HSCR risk ~4-fold [7] and ~130-fold [22], respectively. 

Influence of the genetic background on the expressivity of ENS defects has been reported not only for mouse models of HSCR (with loss-of-function mutation of *Sox10*, *Ednrb*, or *Ret*) [23,24,25] but also for other ENS disorders like intestinal neuronal dysplasia (with loss-of-function mutation of *Tlx2* or *Kif26a*) [26,27]. Almost all these studies tested the C57BL/6 background (either 6N or 6J substrains), which was compared to other strains—like BALB/c, C3Fe, and 129S—that differed as a function of the study. The consensus emanating from these studies is that the C57BL/6 background is the most susceptible to develop ENS defects. Yet, these studies also revealed that this is not the case for all neural crest derivatives. The opposite was seen for neural crest-derived melanocytes in *Sox10*-mutant mice, which presented a more severe pigmentation defect in the C3Fe background compared to the C57BL/6 background [25]. A similar disconnection between ENS and melanocyte defects was also reported in a study investigating the impact of the genetic background on the phenotype of *Ednrb*-mutant rats [28]. 

The *Holstein* mouse model of HSCR is issued from a pigmentation-based forward genetic screen aimed at identifying neurocristopathy-associated loci in FVB/N mice [29]. This screen was based on the transgenic rescue of the albino mutation in the *tyrosinase* gene (*Tyr*) [30,31,32], an approach that can result in non-uniform patterns of pigmentation when the rescuing *Tyr* minigene is inserted in a neural crest-relevant locus [29]. *Holstein* mice display a fully penetrant recessive phenotype combining unpigmented fur with aganglionic megacolon [29,33]. This phenotype is due to insertional mutation of a CTCF insulator element upstream of *Collagen-6 alpha-4* (*Col6a4*) on Chr.9, leading to neural crest-specific upregulation of this gene [33]. The resulting increase of total collagen VI protein levels interferes with ENS progenitor migration in homozygous *Holstein* embryos (*Hol^Tg/Tg^*) [33]. Our analysis of the most distal ENS-containing colon of patients with short-segment HSCR further revealed that collagen VI protein levels are especially elevated in children also having trisomy 21 [33], consistent with the presence of two collagen VI genes (*COL6A1* and *COL6A2*) in a region of Chr.21 that increases HSCR risk when present in 3 copies [34].

Using this model of trisomy 21-associated HSCR (*Hol^Tg/Tg^*) and other models for male-biased (*TashT^Tg/Tg^*) [35,36] and *EDNRB* mutation-associated (*Ednrb^s-l/s-l^*) [37] HSCR, we recently developed a new therapy for HSCR allowing us to generate a new ENS from tissue-resident ENS progenitors in the otherwise aganglionic colon [38]. Acute post-natal administration of the potent neurotrophic factor GDNF via rectal enemas was found to be sufficient not only for inducing ENS ganglia with both neurons and glia but also for globally improving colon structure and function, thereby preventing premature megacolon-associated death of a significant number of treated animals [38]. When exposed to GDNF *in vitro*, cultured explants of aganglionic bowel from children with HSCR also developed new neurons [38]. While the origin of GDNF-induced ENS ganglia is not yet fully understood, about a third of induced neural cells were found to arise from Schwann cells in extrinsic nerves [38], which are more abundant than normal in the aganglionic colon of both mice [24,33,35] and humans [39]. 

In the current study, we verified if (1) the C57BL/6N background could specifically worsen the ENS defect of *Hol^Tg/Tg^* mice as previously reported for other mouse models of HSCR and if (2) this could also impact response to GDNF enemas. Surprisingly, the ENS phenotype of *Hol^Tg/Tg^* mice maintained in the C57BL/6N background (hereafter referred to as *Hol^Tg/Tg^[BL6]*) was found to be less severe than for *Hol^Tg/Tg^* mice maintained in their original FVB/N background (hereafter referred to as *Hol^Tg/Tg^[FVB]*). Accordingly, the outcome of GDNF enemas in terms of resulting ENS density appeared better for *Hol^Tg/Tg^[BL6]* than for *Hol^Tg/Tg^[FVB]* mice. The change of genetic background specifically impacted the ENS, leaving the melanocyte defect of *Hol^Tg/Tg^* unaffected.

## 2. Results

### 2.1. Increased Survival of Hol^Tg/Tg^[BL6] Mice Compared to Hol^Tg/Tg^[FVB] Mice

To determine if changing the genetic background from FVB/N to C57BL/6N could influence the phenotype of *Holstein* mice, heterozygous *Hol^Tg/+^[FVB]* mice were backcrossed with C57BL/6N mice for nine generations. As previously reported for *Hol^Tg/+^[FVB]* mice [33], the resulting *Hol^Tg/+^[BL6]* presented large areas of unpigmented fur covering about half of their body (44.2 ± 2.2%; Figure 1a,b). Intercrosses of *Hol^Tg/+^[BL6]* mice yielded the expected Mendelian ratio of homozygous *Hol^Tg/Tg^[BL6]* mice, which were easily recognizable by their almost entirely unpigmented fur (Figure 1a,b) as previously described for *Hol^Tg/Tg^[FVB]* mice [33].

Yet, *Hol^Tg/Tg^[BL6]* mice markedly differed from *Hol^Tg/Tg^[FVB]* mice in terms of survival rate. Close monitoring of both colonies during the same period revealed an 8.2-fold increase in median survival age of *Hol^Tg/Tg^[BL6]* mice (169 days; interval of 28–417 days) compared to *Hol^Tg/Tg^[FVB]* mice (20.5 days; interval of 14–56 days) (Figure 1c). Regardless of genetic background, all dying *Hol^Tg/Tg^* mice exhibited typical megacolon-associated symptoms including abdominal distention, growth delay/weight loss, hunched posture, and ruffled fur. Accordingly, all of these mice had a narrow distal colon and enlarged mid/proximal colon (Figure 1d). 

The increased survival of *Hol^Tg/Tg^[BL6]* mice further allowed us to test their fertility. All tested breeding pairs (*n* = 21) were fertile, producing an average of 6.2 ± 2.1 pups per litter. However, 43% (9/21) of reproductive *Hol^Tg/Tg^[BL6]* females eventually died of dystocia. At necropsy, all these females displayed fecal impaction, which appeared to compress the vagina. Altogether, these observations are consistent with the notion that the C57BL/6N genetic background specifically decreases the severity of the ENS defect of *Hol^Tg/Tg^* mice, without influencing their pigmentation defect. 

### 2.2. Short-Segment Aganglionosis and Colonic Dysmotility in Hol^Tg/Tg^[BL6] Mice 

To confirm that aganglionosis was the underlying cause of megacolon/fecal impaction in *Hol^Tg/Tg^[BL6]* mice, we examined their myenteric plexus at P20 via staining for acetylcholinesterase (AchE) activity (Figure 2a). As previously observed in *Hol^Tg/Tg^[FVB]* mice [33], the rectum of *Hol^Tg/Tg^[BL6]* mice is completely devoid of myenteric ganglia which are instead replaced by hypertrophic extrinsic nerve fibers (Figure 2a,b). This aganglionic segment is preceded by a transition zone with markedly decreased ENS density (25.4 ± 3.4% in mutants vs. 38.7 ± 1.1% in controls), while no overt differences are noted in the proximal colon (Figure 2a,b). In accordance with the increased survival in the C57BL/6N background (Figure 1c), we further found that the length of ENS-covered colon is longer for *Hol^Tg/Tg^[BL6]* mice (86.6 ± 1.2%; Figure 2c) than in *Hol^Tg/Tg^[FVB]* mice (74.2 ± 2.2%, [33]). Interestingly, this analysis also revealed that the majority of *Hol^Tg/Tg^[BL6]* mice had reached what we previously described as the minimal length of ENS innervation necessary to avoid blockage in the FVB/N background (between 78–84%, [35,36]; see dashed lines in Figure 1c), suggesting that the same threshold level applies in the C57BL/6N background.

To verify the functional impact of short-segment aganglionosis in *Hol^Tg/Tg^[BL6]* mice, we analyzed colonic motility at P20 via the bead latency test (Figure 2d). Although most *Hol^Tg/Tg^[BL6]* mice were able to expel a rectally inserted glass bead, it took them much longer than for control C57BL/6N mice (17.1 ± 2.8 min vs. 5.5 ± 1.0 min, respectively). *Hol^Tg/Tg^[BL6]* mice that have not expelled the bead during the 60-min test were only rarely observed, whereas this was the norm for *Hol^Tg/Tg^[FVB]* mice [38]. Direct comparison with the length of the ENS-covered colon revealed a robust inverse correlation between time to expel the bead and the extent of aganglionosis (Figure 2e). All these data confirm that the ENS defect is less severe in *Hol^Tg/Tg^[BL6]* than in *Hol^Tg/Tg^[FVB]* mice.

### 2.3. Col6a4 Overexpression Is Less Extensive in Hol^Tg/Tg^[BL6] than in Hol^Tg/Tg^[FVB] Mice

Aganglionosis in *Hol^Tg/Tg^[FVB]* mice is due to transgene insertion-induced upregulation of *Col6a4* and accompanying excessive, anti-migratory, secretion of collagen VI microfibrils by neural crest-derived ENS progenitors [33]. To determine the impact of the C57BL/6N background on this pathological mechanism, we first analyzed the extent of colonization by ENS progenitors via whole-mount immunofluorescence staining of βIII-tubulin. As previously reported for the *Hol^Tg/Tg^[FVB]* line [33], colonization by ENS progenitors was found to be less extensive in the colon of *Hol^Tg/Tg^[BL6]* embryos in comparison to WT controls (Figure 3). However, at both analyzed stages (e12.5 and e15.5), this phenotype again appears less severe in the C57BL/6N background (83% and 77% of WT levels at e12.5 and e15.5, respectively; Figure 3) compared to the FVB/N background (52% and 62% of WT levels at e12.5 and e15.5, respectively [33]).

We next sought to analyze *Col6a4* expression levels by RT-qPCR, specifically in ENS progenitors. For this analysis, we thus introduced the *Gata4p[5kb]-RFP* (*G4-RFP*) transgene [40] in the *Hol^Tg/Tg^[BL6]* background by breeding—this transgene allowing to recover of RFP-positive ENS progenitors by fluorescence-activated cell sorting (FACS) [33,41]. Then, using e12.5 ENS progenitors recovered from these *Hol^Tg/Tg^[BL6];*
*G4-RFP* embryos as well as from *Hol^Tg/Tg^[FVB];*
*G4-RFP* embryos [33], we measured mRNA levels of *Col6a4* and both of the collagen VI gene isoforms required for the production of collagen VI microfibrils (*Col6a1* and *Col6a2*) [42]. In line with all other phenotypic data, our comparative analysis revealed a specific 2-fold decrease of *Col6a4* expression levels in the C57BL/6N background compared to the FVB/N background, without any impact on *Col6a1* and *Col6a2* levels (Figure 4a). Using immunofluorescence, we further confirmed that this specific variation of *Col6a4* expression levels similarly impacted the overall production of collagen VI microfibrils in e12.5 guts. Compared to respective WT controls, the immunofluorescence signal of collagen VI was found to be either doubled or unaffected in small intestines and cecum from *Hol^Tg/Tg^[BL6]* embryos (Figure 4b–e), whereas it was previously found to be tripled in all corresponding regions from *Hol^Tg/Tg^[FVB]* embryos [33]. These results strongly suggest that the *Holstein* ENS defect differs in FVB/N and C57BL/6N backgrounds because associated genetic differences are modulating the impact of the insertional mutation on *Col6a4* gene expression.

### 2.4. GDNF Enemas Restore Nearly Normal ENS Density and Function in Hol^Tg/Tg^[BL6] Mice

Having recently discovered that GDNF enemas can induce a new ENS in the otherwise aganglionic colon of several mouse models of HSCR (including *Hol^Tg/Tg^[FVB]*) [38], we wondered if the genetic background might influence the response to this treatment. To address this question, we used the exact same experimental approach as before and administered GDNF enemas (10 µL at 1 µg/µL) to *Hol^Tg/Tg^[BL6]* pups once daily between P4 and P8. Then, we analyzed the response to GDNF treatment via whole-mount immunofluorescence staining of HuC/D+ myenteric neurons at P20. As previously observed in *Hol^Tg/Tg^[FVB]* animals [38], GDNF treatment markedly increased the number of HuC/D+ myenteric neurons in *Hol^Tg/Tg^[BL6]* mice, reaching WT-like levels in ENS-containing colon (both proximal and distal) and about two-thirds of WT levels in the otherwise aganglionic rectum (Figure 5a and Appendix A). The magnitude of the response to GDNF enemas thus appears greater in *Hol^Tg/Tg^[BL6]* mice than in *Hol^Tg/Tg^[FVB]* mice, in which density of GDNF-induced ENS only reached about a third of WT levels in otherwise aganglionic distal colon [38].

To confirm the functionality of GDNF-induced myenteric ganglia, we analyzed colonic motility using the bead latency test. In contrast to untreated *Hol^Tg/Tg^[BL6]* mice for which one third (2/6) did not expel the bead before the end of the assay (average of 25.6 ± 2.4 min for the entire group), all GDNF-treated *Hol^Tg/Tg^[BL6]* mice did expel the bead (average of 13.2 ± 3.0 min) although not as quick as for control C57BL/6N mice (average of 3.2 ± 0.6 min) (Figure 5b). Strikingly, the time needed for GDNF-treated *Hol^Tg/Tg^[BL6]* mice to expel the bead was also found to be inversely correlated with the percentage of ENS density in the rectum (Figure 5c). This set of data suggests that GDNF-based correction of *Holstein*-associated ENS defect is more efficient in the C57BL/6N background than in the FVB/N background. 

## 3. Discussion

In the current study, we first wanted to determine if the genetic background could influence the expressivity of aganglionic megacolon in mice bearing the *Holstein* insertional mutation, as observed for human HSCR. Transfer of the *Holstein* mutation from the FVB/N to the C57BL/6N background specifically decreased the severity of the collagen VI-dependent ENS defect of *Hol^Tg/Tg^* mice, without influencing its associated pigmentation defect. On one hand, this finding thus rules out the possibility that the C57BL/6 background is a general sensitized background for ENS defects, as previously suggested by similar analyses of mice mutated for *Sox10*, *Ednrb*, *Tlx2*, or *Kif26a* [23,25,26,27]. On the other hand, the observed genetic background-induced disconnection in severity between these two neural crest-related phenotypes replicates prior findings made in rodents bearing mutation of *Sox10* or *Ednrb* [25,28]. These observations are consistent with the notion that the formation of the ENS is more sensitive to genetic variations than the melanocyte lineage. This most likely has something to do with the much greater number of genes required for building a complex structure like the ENS as opposed to the generation of a single cell lineage (melanocyte in this case). The corollary of this is that the greater genetic complexity associated with ENS formation obviously provides a richer source of phenotypically impactful variations.

Our *Col6a4* RT-qPCR and collagen VI immunofluorescence data suggest that local genomic changes around the *Holstein* transgene insertion site might be one reason for the observed genetic background effect. The *Holstein* transgenic insertion is believed to perturb the insulation activity of a region enriched in CTCF binding motifs separating *Col6a4* from a neighbor region highly transcribed in neural crest cells [33]. In agreement with this possibility, close examination of this CTCF-enriched region using the Ensembl Genome Browser revealed the presence of an indel variant immediately downstream of a CTCF ChIP-seq peak (Appendix A). As confirmed by Sanger sequencing of DNA amplicons from our *Holstein* mouse colonies, the corresponding 12-bp sequence is present in the FVB/N background but absent in the C57BL/6N background (Appendix A). The functional impact of this particular indel variant is currently unknown. However, previous studies in other pathological contexts suggest that variations flanking CTCF binding sites can influence the binding of CTCF proteins and thereby insulation activity of corresponding regions [43,44]. 

Variation in the pathological secretion of collagen VI also provides a plausible explanation as to why GDNF treatments appear more efficient in *Hol^Tg/Tg^[BL6]* mice compared to *Hol^Tg/Tg^[FVB]* mice. Indeed, the extracellular matrix is a major determinant of GDNF diffusion and activity [45,46]. Accordingly, levels of collagen VI in the diseased segment might affect both the distribution of rectally administered GDNF and the subsequent response of targeted ENS progenitors. Although the impact on GDNF diffusion in the colon is currently unknown, we do know that GDNF-induced migration of ENS progenitors from gut explants is more efficient when collagen VI levels are reduced [33]. Our data thus add credence to the growing belief that modulation of the extracellular matrix around ENS progenitors might have therapeutic value for enteric neuropathies [2,47]. 

## 4. Materials and Methods

### 4.1. Animals

*Holstein* (*Hol^Tg/Tg^[FVB])* and *Gata4p[5kb]-RFP* (*G4-RFP*) single and double transgenics in the FVB/N background were as previously described [33,40]. Both transgenic alleles were transferred on the C57BL/6N background via consecutive backcrossing over 5 (for *G4-RFP*) to 9 (for *Holstein*) generations and resulting single transgenic lines were then intercrossed to generate double transgenics. Genotyping of adult *Hol^Tg/Tg^[BL6]* animals was made by visual inspection of coat color whereas genotyping of *Holstein* and *G4-RFP* alleles in embryonic tissues was performed by standard PCR using primers listed in Appendix A. For embryo analyses, mice were mated during the night and noon on the day of vaginal plug detection was designated as e0.5. Euthanasia of pregnant dams or mice used for postnatal analyses was performed by CO_2_ inhalation following isoflurane-mediated anesthesia. Clinical-grade GDNF (Medgenesis Therapeutix Inc, Victoria, BC, Canada) was administered to some *Hol^Tg/Tg^[BL6]* pups via rectal enemas (10 µL of a 1 µg/µL solution in phosphate-buffered saline) once a day for 5 consecutive days from P4 to P8, as previously described [38].

### 4.2. Tissue Staining and Imaging

For postnatal tissues, whole colons were dissected from P20 mice, cut longitudinally along the mesentery, and fixed in 4% paraformaldehyde (PFA) overnight at 4 °C. Muscle layers and associated myenteric plexus were then microdissected from mucosa/submucosa and subsequently stained for acetylcholinesterase activity [33,48] or via immunofluorescence [38], as previously described. For embryonic tissues, whole intestines were dissected from e12.5 and e15.5 embryos, fixed in 4% PFA for 1 h at room temperature, and then processed for immunofluorescence staining, as previously described [33]. Details about all antibodies used in this study are listed in Appendix A. All immunofluorescence images were acquired with 20×/60× objectives on a Nikon A1R confocal unit. To quantify immunofluorescence signal, relevant images were acquired using the exact same settings, and mean fluorescence intensity in candela/μm^2^ was determined within 20× fields of view using the Image J software, as previously described [33]. Images of acetylcholinesterase staining were acquired with a Leica DFC 495 camera mounted on a Leica M205 FA stereomicroscope (Leica Microsystems Canada, Vaughan, ON, Canada).

### 4.3. Bead Latency Test

Distal colonic motility was analyzed using an in vivo bead expulsion assay, as previously described [38,49]. Briefly, a 2-mm glass bead (Sigma-Aldrich, St. Louis, MO, USA) was inserted using a probe over a distance of 0.5 cm from the anus of overnight-fasted mice, under isoflurane anesthesia. Mice were then individually isolated in their cage without access to food and water, and colonic transit was determined by monitoring the time required for bead expulsion after insertion. The maximum time allowed for bead expulsion was set at either 30 or 60 min.

### 4.4. Fluorescence-Activated Cell Sorting (FACS) and RT-qPCR

Whole intestines were dissected from e12.5 embryos obtained from *Hol^Tg/+^[FVB]*; *G4-RFP* or *Hol^Tg/+^[BL6]*; *G4-RFP* intercrosses and individually dissociated at 37 °C using a cocktail of collagenase (0.4 mg/mL; Sigma C2674), dispase II (1.3 mg/mL; Life Technologies 17105-041, Carlsbad, CA, USA) and DNAse I (0.5 mg/mL; Sigma DN25), as previously described [33]. Single RFP-positive cells (~10,000 per intestine) were recovered from each preparation using a FACSJazz cell sorter (BD Biosciences, San Jose, CA, USA) and then kept frozen at −80 °C. Following PCR-based genotyping of embryo heads, relevant samples were individually processed for RNA extraction using the RNAeasy Plus purification mini kit (Qiagen, Germantown, MD, USA) and reverse transcription (with 50 ng of total RNA) using the Superscript III kit (Invitrogen, St. Louis, MO, USA), in accordance with manufacturers’ instructions. RT-qPCR analyses were performed with the Ssofast EvaGreen Supermix and C1000 Touch thermal cycler (BioRad, Hercules, CA, USA), in accordance with the manufacturer’s protocol. PCR consisted of 35 cycles of 20 s at 95 °C, 40 s at 60 °C, and 60 s at 72 °C. Quantitative gene expression relative to *Gapdh* was calculated by the 2−ΔΔCt method [50]. All primers used for RT-qPCR are described in Appendix A.

### 4.5. Statistical Analysis

For all quantitative analyses, data are expressed as the mean ± standard error of the mean (SEM). The number of independent biological replicates (*n*) is indicated either in the figure or accompanying legend. The significance of differences was determined using relevant tests in GraphPad Prism, as indicated in figure legends. *p* values below 0.05 were considered statistically significant.

## Figures and Tables

**Figure 1 ijms-22-13140-f001:**
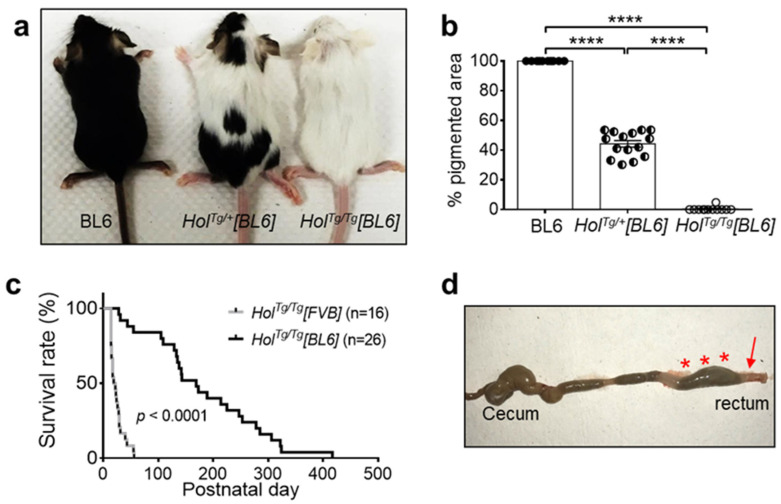
The C57BL/6N genetic background delays megacolon-associated death in *Hol^Tg/Tg^* mice. (**a**,**b**) Allele dosage-dependent decrease of pigmentation in P20 *Hol^Tg/Tg^[BL6]* mice. **** *p* < 0.0001, one-way ANOVA with post-hoc Sidak’s test. (**c**) Comparison of survival rates between *Hol^Tg/Tg^[BL6]* and *Hol^Tg/Tg^[FVB]* mice. **** *p* < 0.0001, Mantel–Cox test. (**d**) *Hol^Tg/Tg^[BL6]* mice die from complications of megacolon, as evidenced in this euthanized P31 animal by the blockage in the distal colon and rectum (arrow) and accumulation of fecal material in more proximal regions (asterisks).

**Figure 2 ijms-22-13140-f002:**
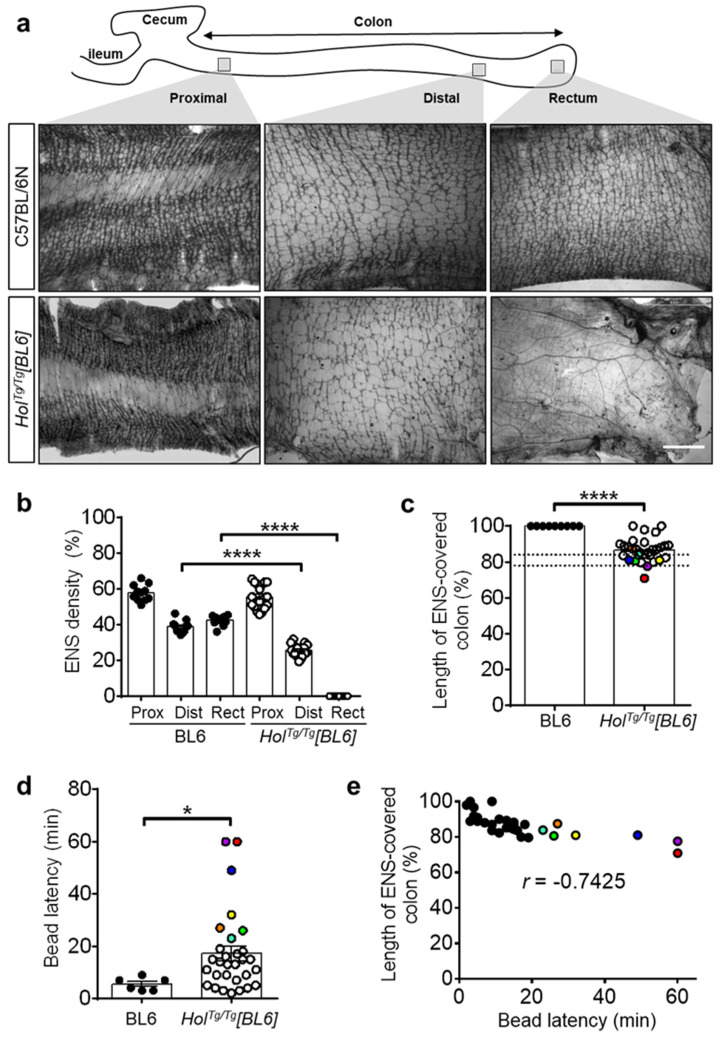
*Hol^Tg/Tg^[BL6]* mice display short-segment aganglionosis and colonic dysmotility. (**a**) Staining of AChE activity in muscle strips from P20 mice, evidencing hypoganglionosis in the distal colon and aganglionosis combined to an overabundance of extrinsic nerve fibers in the rectum of *Hol^Tg/Tg^[BL6]*. Scale bar, 1 mm. (**b**) Quantitative analysis of the area occupied by the myenteric plexus (expressed in % of the image area), using images such as those displayed in (**a**). Each value is a field a view, for a minimum of 3 fields of view per region (*n* = 3–5 animals per group). Values for the aganglionic rectum of *Hol^Tg/Tg^[BL6]* mice were set at 0% by default, to avoid the confounder extrinsic nerves. **** *p* < 0.0001, one-way ANOVA with post-hoc Sidak’s test. (**c**) Quantitative analysis of the length of ENS-covered colon (in % of total colon length) in P20 mice. The dashed lines refer to the previously described threshold level interval beyond which megacolon is less likely to occur in FVB/N mice [35,36]. **** *p* < 0.0001, two-tailed Student’s *t*-test. (**d**) In vivo analysis of colonic motility in P20 mice using the bead latency test. Time to expel the glass bead after rectal insertion was capped at 60 min to simplify the analysis without impacting statistical significance. * *p* < 0.05, two-tailed Mann–Whitney U test. (**e**) Correlation between length of ENS-covered colon and time to expel the bead. Colored symbols were attributed to all *Hol^Tg/Tg^[BL6]* mice with latency time above average in panel d and used to highlight the same mice in panels c and e. *r*, Pearson’s correlation coefficient.

**Figure 3 ijms-22-13140-f003:**
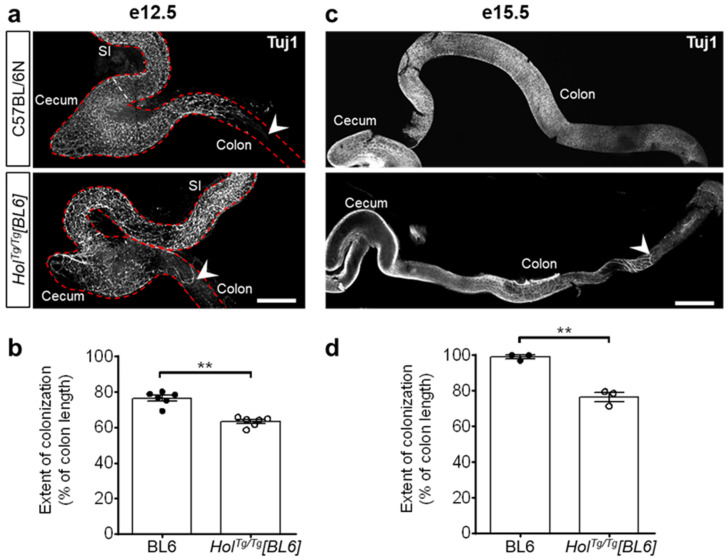
Colonization by ENS progenitors has delayed in the colon of *Hol^Tg/Tg^[BL6]* embryos. (**a**,**c**) Immunofluorescence staining of βIII-Tubulin (Tuj1) in whole-mount preparations of embryonic guts at e12.5 (**a**) and e15.5 (**c**), with migratory front indicated by arrowheads. Scale bar, 400 µm (**a**) and 700 µm (**c**). (**b**,**d**) Quantification of extent of colonization by ENS progenitors (expressed in % of total colon length) at e12.5 (**b**) and e15.5 (**d**), using images such as those displayed in panels a and c. ** *p* < 0.01, two-tailed Student’s *t*-test.

**Figure 4 ijms-22-13140-f004:**
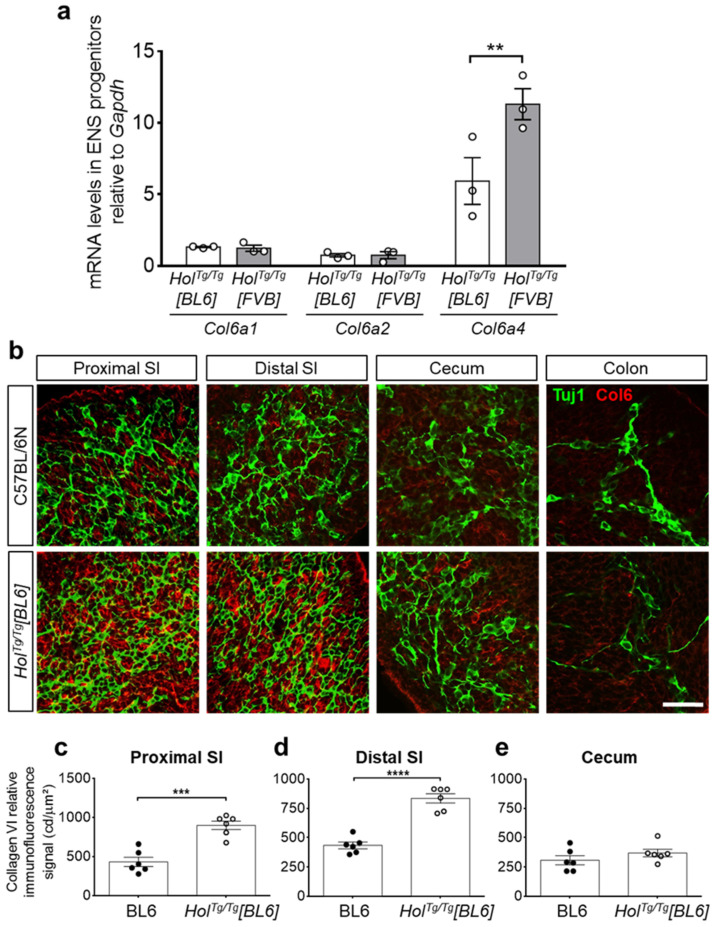
The increased gene expression of *Col6a4* and secretion of collagen VI microfibrils are both lower in *Hol^Tg/Tg^[BL6]* embryos than in *Hol^Tg/Tg^[FVB]* embryos. (**a**) RT-qPCR analysis of *Col6a1*, *Col6a2*, and *Col6a4* in e12.5 ENS progenitors recovered by FACS from *Hol^Tg/Tg^[BL6];*
*G4-RFP* and *Hol^Tg/Tg^[FVB];*
*G4-RFP* embryos. ** *p* < 0.01, two-way ANOVA with post-hoc Sidak’s test. (**b**) Double immunofluorescence staining of collagen VI microfibrils (Col6, red) and βIII-tubulin+ neuronal progenitors (Tuj1, green) in e12.5 embryonic guts from control C57BL/6 and mutant *Hol^Tg/Tg^[BL6]* embryos. SI, small intestine. Scale bar, 20 µm. (**c**–**e**) Quantitative analysis of collagen VI immunofluorescence signal (in candella [cd] per µm^2^) in e12.5 C57BL/6 and *Hol^Tg/Tg^[BL6]* guts, using images such as those displayed in panel b. *** *p* < 0.001, **** *p* < 0.0001, two-way ANOVA with post-hoc Sidak’s test.

**Figure 5 ijms-22-13140-f005:**
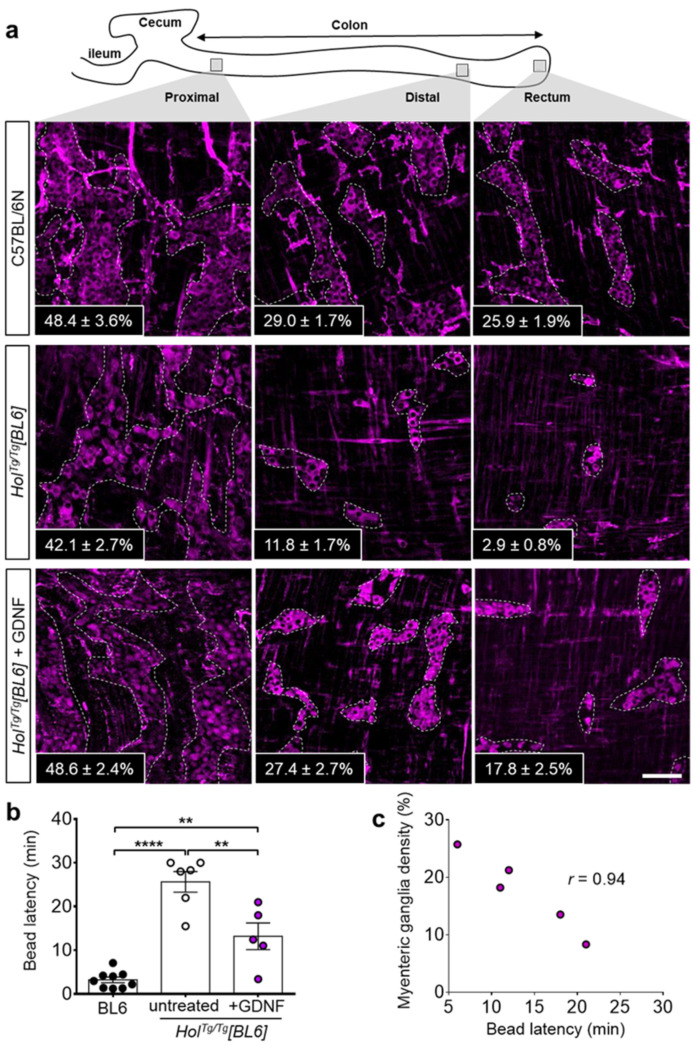
GDNF-induced neurogenesis in the colon of *Hol^Tg/Tg^[BL6]* pups. (**a**) Immunofluorescence staining of HuC/D+ neurons in the myenteric plexus of P20 C57BL/6 controls and *Hol^Tg/Tg^[BL6]* mutants treated or not with GDNF between P4–P8. Insets refer to the average percentage of surface area occupied by HuC/D+ myenteric ganglia in 3–4 fields of view per corresponding region (*n* = 3–5 mice per group, as detailed in Appendix A). Scale bar, 100 µm. (**b**) In vivo analysis of colonic motility in P20 mice using the bead latency test. Time to expel the glass bead after rectal insertion was capped at 30 min to simplify the analysis without impacting statistical significance. ** *p* < 0.01, **** *p* < 0.0001, two-way ANOVA with post-hoc Sidak’s test. (**c**) Correlation between myenteric ganglia density and time to expel the bead for P20 *Hol^Tg/Tg^[BL6]* mice treated with GDNF between P4–P8.

## Data Availability

All data are included in the current article and associated Appendix A.

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
