# Peer review of "Genetic Background Influences Severity of Colonic Aganglionosis and Response to GDNF Enemas in the Holstein Mouse Model of Hirschsprung Disease"

_ijms, 2021, doi:10.3390/ijms222313140_

Round 1

Reviewer 1 Report

This is a study to address if genetic background affect the penetrance and severity of HSCR gene (in this study the Col6a4). The authors also test if genetic background affect the efficacy of GDNF enemas in the regeneration of the ENS. The results confirmed that the severity of colonic aganglionosis is indeed affect by genetic background and the responsiveness of GDNF enemas. The authors also suggest that the 12-bp sequence which is present in the FVB/N background but absent in the C57BL/6N background may be the reason for the genetic background influence.  The impact of genetic background to HSCR is not new and it has been known in human patients and in animal models of HSCR.

Major comment:

(1) As the transgene insertion and the CFTCF binding sites are quite close, the 12-bp sequence is very close to the CTCF bind site. It is not clear how far apart from the transgene insertion site and the CTCF binding sites and the 12-bp sequence.  Author on cross to different genetic background for 9 generations. It is not clear if there is recombination between these three sites. Author should perform PCR sequencing to confirm the recombination amongst these sites.

Author Response

Thank you for the time spent on reviewing our manuscript. In response to your comment, we did sequence PCR products of the CTCF region and accompanying 12bp indel variant amplified from HolTg/Tg[B6] and HolTg/Tg[FVB] genomic DNA. Corresponding sequence alignment was presented in Fig. S2, but we agree that the source of DNA was unclear from this Figure panel (it was indicated in the Figure legend only). We have now amended this Figure to make both the DNA source and presence of CTCF region clearer.

Reviewer 2 Report

-In this manuscript Soret et al., describe how the different genetic background of the Holstein mouse model of Hirschprung’s disease affects the enteric nervous system development. The authors demonstrate how the efficacy of GDNF treatment was improved in Holstein strain with C57BL/6N background. Greater understanding of the role of genetic background in enteric nervous system formation, emphasizing the complex nature of the extracellular matrix, specifically collagen type 6, is a great value to the developmental genetics field. The paper is clearly written, technically sounds and it is suitable for publication in IJMS.

Author Response

Thank you for the time spent on reviewing our manuscript. We are glad that you liked it!

Round 2

Reviewer 1 Report

Authors back-cross a HSCR mice to test the influence of genetic background on HSCR expressivity. The data indeed show a difference of HSCR phenotypes in C67BL6 and FVB background. My main concern is the novelty of the finding. Influence of genetic background on HSCR phenotype is not new in both human and animal model (especially in rodents). The interesting finding is on the 12-bp sequence that is present in FVB bot not in C57BL6 mice, could be the reason of the difference of HSCR phenotype in two different genetic backgrounds.

Major comments:-

But, the separation between this 12-bp and the Tg insertion site and the 153bp deletion is not reported and whether there is recombination of the Tg and the 12-bp sequence has happened in just heterozygous HolTg/+[FVB] mice back-crossing with C57BL/6N mice for just nine generations. Author should consider experiment to show if there is indeed recombination between the Tg and the 12-bp sequence.

Author Response

We really do not see what else can be done other than what is already presented in Fig.S2. The sequence alignment of PCR products in panel B clearly shows that the 12bp sequence near the CTCF site is present in HolTg/Tg[FVB] but not in HolTg/Tg[B6] mice, which by definition are both bearing the mutagenic transgene insertion. The only possible interpretation of this data is that the CTCF-associated 12bp indel variant did not segregate with the mutagenic transgene (at ~12kb on the centromeric side) at some point during backcrossing.